# Comparative Study of the Role of Interepithelial Mucosal Mast Cells in the Context of Intestinal Adenoma-Carcinoma Progression

**DOI:** 10.3390/cancers14092248

**Published:** 2022-04-30

**Authors:** Tanja Groll, Miguel Silva, Rim Sabrina Jahan Sarker, Markus Tschurtschenthaler, Theresa Schnalzger, Carolin Mogler, Daniela Denk, Sebastian Schölch, Barbara U. Schraml, Jürgen Ruland, Roland Rad, Dieter Saur, Wilko Weichert, Moritz Jesinghaus, Kaspar Matiasek, Katja Steiger

**Affiliations:** 1Institute of Pathology, School of Medicine, Technical University of Munich, 81675 Munich, Germany; tanja.groll@tum.de (T.G.); miguel.silva@tum.de (M.S.); sabrina.sarker@tum.de (R.S.J.S.); carolin.mogler@tum.de (C.M.); daniela.denk@tum.de (D.D.); wilko.weichert@tum.de (W.W.); moritz.jesinghaus@uni-marburg.de (M.J.); 2Comparative Experimental Pathology (CEP), School of Medicine, Technical University of Munich, 81675 Munich, Germany; 3Center for Clinical Veterinary Medicine, Institute of Veterinary Pathology, Ludwig-Maximilians-Universitaet (LMU), 80539 Munich, Germany; kaspar.matiasek@neuropathologie.de; 4Department of Medicine II, Klinikum Rechts der Isar, School of Medicine, Technical University of Munich, 81675 Munich, Germany; markus.tschurtschenthaler@tum.de (M.T.); roland.rad@tum.de (R.R.); dieter.saur@tum.de (D.S.); 5German Cancer Consortium (DKTK), German Cancer Research Center (DKFZ), Partner Site Munich, 81675 Munich, Germany; j.ruland@tum.de; 6TranslaTUM, Center for Translational Cancer Research, Technical University of Munich, 81675 Munich, Germany; theresa.schnalzger@tum.de; 7Institute of Translational Cancer Research and Experimental Cancer Therapy, Klinikum Rechts der Isar, School of Medicine, Technical University of Munich, 81675 Munich, Germany; 8Institute of Clinical Chemistry and Pathobiochemistry, School of Medicine, Technical University of Munich, 81675 Munich, Germany; 9JCCU Translational Surgical Oncology (A430), German Cancer Research Center (DKFZ), 69120 Heidelberg, Germany; s.schoelch@dkfz.de; 10DKFZ-Hector Cancer Institute at University Medical Center Mannheim, 68167 Mannheim, Germany; 11Department of Surgery, Medical Faculty Mannheim, Heidelberg University, 68167 Mannheim, Germany; 12Walter Brendel Centre of Experimental Medicine, University Hospital, LMU Munich, 82152 Planegg-Martinsried, Germany; barbara.schraml@bmc.med.lmu.de; 13Biomedical Center (BMC), Institute for Cardiovascular Physiology and Pathophysiology, Faculty of Medicine, LMU Munich, 82152 Planegg-Martinsried, Germany; 14Institute of Molecular Oncology and Functional Genomics, School of Medicine, Technical University of Munich, 81675 Munich, Germany; 15Institute of Pathology, University Hospital Marburg, 35043 Marburg, Germany

**Keywords:** tumor microenvironment, colorectal cancer, interepithelial mucosal mast cells, adenoma-carcinoma sequence, human, genetically engineered mouse models

## Abstract

**Simple Summary:**

The role of mast cells in the tumor microenvironment (TME) remains controversial but has become increasingly evident and explored as a possible therapeutic target. In this study, we investigated the underexplored mast cell heterogeneity of intestinal cancer by applying standardized mast cell subtyping during adenoma-carcinoma progression. We immunohistochemically evaluated and scored the occurrence of interepithelial mucosal mast cells (ieMMCs) in tumors of frequently employed genetically engineered mouse models and in human colonic adenomas and carcinomas. We found a decrease of ieMMCs from colonic low-grade adenomas to carcinomas. Moreover, mouse models based on altered Wnt signaling showed higher ieMMC scores than models based on altered MAPK signaling. Our descriptive study indicates that ieMMCs play a special role in the molecular TME related to adenoma-carcinoma progression. Furthermore, it emphasizes the need for adequate immunohistochemical methodology and experimental setup when investigating the functional role of mast cell populations of the TME.

**Abstract:**

Mast cells (MCs) are crucial players in the relationship between the tumor microenvironment (TME) and cancer cells and have been shown to influence angiogenesis and progression of human colorectal cancer (CRC). However, the role of MCs in the TME is controversially discussed as either pro- or anti-tumorigenic. Genetically engineered mouse models (GEMMs) are the most frequently used in vivo models for human CRC research. In the murine intestine there are at least three different MC subtypes: interepithelial mucosal mast cells (ieMMCs), lamina proprial mucosal mast cells (lpMMCs) and connective tissue mast cells (CTMCs). Interepithelial mucosal mast cells (ieMMCs) in (pre-)neoplastic intestinal formalin-fixed paraffin-embedded (FFPE) specimens of mouse models (total lesions *n* = 274) and human patients (*n* = 104) were immunohistochemically identified and semiquantitatively scored. Scores were analyzed along the adenoma-carcinoma sequence in humans and 12 GEMMs of small and large intestinal cancer. The presence of ieMMCs was a common finding in intestinal adenomas and carcinomas in mice and humans. The number of ieMMCs decreased in the course of colonic adenoma-carcinoma sequence in both species (*p* < 0.001). However, this dynamic cellular state was not observed for small intestinal murine tumors. Furthermore, ieMMC scores were higher in GEMMs with altered Wnt signaling (active β-catenin) than in GEMMs with altered MAPK signaling and wildtypes (WT). In conclusion, we hypothesize that, besides stromal MCs (lpMMCs/CTMCs), particularly the ieMMC subset is important for onset and progression of intestinal neoplasia and may interact with the adjacent neoplastic epithelial cells in dependence on the molecular environment. Moreover, our study indicates the need for adequate GEMMs for the investigation of the intestinal immunologic TME.

## 1. Introduction

Worldwide, colorectal cancer (CRC) is the third-most frequent cancer type and the second most common cause of cancer-related deaths [1]. It is well known that mast cells (MCs) are important players in the immunologic microenvironment of intestinal tumors [2]. However, to date, their specific role in the intestinal tumor microenvironment (TME) and their contribution to the molecular intestinal carcinogenesis remain diverse and largely elusive. On one hand, there is evidence for a pro-tumorigenic (i.e., poor prognostic) role [3,4,5,6]. On the other hand, several studies describe an anti-tumorigenic (i.e., favorable prognostic) role of MCs [7,8]. Recent literature suggests evidence of a cellular crosstalk between cancer cells and MCs in the context of intestinal carcinogenesis [9,10,11,12]. Ultimately, an advanced understanding of the role of MCs in CRC will also be beneficial with regard to therapeutic options [13].

Historically, rodent MCs are divided into T-cell independent connective tissue mast cells (CTMCs) and T-cell dependent mucosal MCs (MMCs) [14,15,16]. Vogel et al., 2018 elaborated this classification in the context of intestinal immunopathology by further distinguishing between (smaller) interepithelial mucosal mast cells (ieMMCs) and lamina proprial mucosal mast cells (lpMMCs) and (larger) CTMCs. Using specific expression of murine mast cell protease 1 and 4 (MCPT1 and MCPT4) and cellular topography, it is possible to differentiate between ieMMCs and lpMMCs/CTMCs within the intestine. The authors emphasize the importance of reliably distinguishing these subtypes by means of immunohistochemistry (IHC) in order to adequately reflect the actual mast cell heterogeneity in the GI tract [17]. Mast cells are generally termed as “subepithelial inflammatory cells” of the lamina propria, located below the intestinal epithelium [18,19]. Nevertheless, it has long been known that MCs are able to migrate through the basal lamina in interepithelial spaces [20].

Previous tissue-based studies of intestinal MCs in human CRC either used no MC subtyping (histochemical stainings) [21,22] or IHC-based methodologies to subtype MCs regarding to their protease content as follows: tryptase (MC_T_) [3,5,9,23]; chymase (MC_C_); or tryptase and chymase (MC_TC_) positive (^+^) MCs [7,8,24,25]. Indeed, some authors occasionally mention the occurrence of intestinal MCs in close proximity to the intestinal epithelium [22], but a clear discrimination between ieMMCs and stromal MCs in human CRC has not been carried out so far.

The human intestinal carcinogenesis is generally regarded as a progressive set of events, starting from the onset of pre-malignant adenoma then developing into invasive carcinoma through the sequential acquisition of genetic alterations in tumor suppressor genes (e.g., *APC*) and oncogenes (e.g., *KRAS*) [26,27,28]. Genetically engineered mouse models (GEMMs) are regarded as the gold standard for the in vivo study of CRC due to the high similarity of disease expression and molecular features between mice and humans [29,30,31].

Considering the dominating role of mice as experimentally induced in vivo models of human CRC, we standardized MC classification by subdividing intestinal MC populations as suggested by Vogel et al., 2018 [17] in order to ensure inter-species comparison. An immunohistological, descriptive, semiquantitative scoring method was established to investigate the relevance of MCs, specifically the understudied ieMMC population, in different groups of GEMMs as well as in human CRC patients. Here we investigated the dynamics of ieMMC populations across the adenoma-carcinoma progression [26,27,32] in mice and humans. Furthermore, we analyzed potential differences in the occurrence of ieMMCs in a variety of GEMMs of small and large intestinal neoplasia. If the term “murine” is used in this article, it will refer to the species mouse (*Mus musculus*).

## 2. Materials and Methods

### 2.1. Archival Mouse Tissue Cohorts

In total, 151 formalin-fixed paraffin-embedded (FFPE) blocks were retrospectively analyzed, originating from different GEMMs generated between the years 2009 and 2021. In some cases, blocks contained more than one neoplastic lesion (Swiss rolls), thus overall *n* = 274 murine lesions were investigated. All FFPE samples were obtained from the archive of the Comparative Experimental Pathology (CEP) unit at the Institute of Pathology, Technical University of Munich (TUM), Munich, Germany. FFPE blocks were stored in a cool, dark, and dry archive cabinet in order to avoid a loss of antigenicity. The mouse models assessed in this study (*n* = 12) were either of wildtype (WT) or expressed genetic alterations in one or more genes which affect either a single or various oncogenic pathways commonly altered in CRC [33,34,35,36,37,38,39]. Therefore, they were used as in vivo models for the investigation of intestinal carcinogenesis in previous experiments. Models were grouped according to: (1.) their genetic alterations; (2.) mode of tumor induction (Appendix A).

#### Genetically Engineered Mouse Models

Mouse models were generated or provided by different collaboration partners (M.S., M.T., T.S., S.S., B.U.S., J.R., R.R., D.S.) and experiments were approved by the local animal welfare committees (Regierung von Oberbayern, Munich, Germany, ROB-55.2-2532.Vet_02-17-79 (D.S., M.T.); Regierung von Oberbayern, Munich, Germany, ROB-55.2-2532.Vet_02-15-26, ROB-55.2-2532.Vet_02-14-86 and ROB-55.2-2532.Vet_02-20-9 (B.U.S., J.R., T.S.); Regierungspräsidium Karlsruhe and Landesdirektion Sachsen, Germany, AZ G-188/11 (S.S.); UK Home Office (M.S., R.R.)). Proper genotyping of each mouse and model was assured by the collaboration partners. For details on the employed mouse models (genotypes pseudonymized), affected pathways, and mouse groups, please refer to Appendix A. Intestinal tumors in WT were chemically induced via administration of Azoxymethane (AOM)/Dextran Sodium Sulfate (DSS) [40]. Active β-catenin mouse models (βCAT) were generated by deleting exon 3 of the β-catenin gene via an intestinal Cre-loxP system [38]. *Apc^fl/wt^* [37], *Kras^G12D^* [36], and *Braf^V637E^* [33] mice were crossed with an intestinal epithelial cell-specific Cre driver line [41]. For models with tumor induction via a surgical Adeno-Cre infection [34], animals with an APC; KRAS background were crossed in different combinations harboring genetic alterations in various genes commonly altered in sporadic CRC (“complex” models) [35]. Mouse models with a germline ablation of Interferon-gamma (IFNγ) [42] and GEMMs with a deficiency in the CARD/BCL10/MALT1 (CBM) signaling pathway—in either an immune cell (IC) subset, intestinal epithelial (IE) cells, or full-body KO—were also amongst the investigated GEMMs (Appendix A).

### 2.2. Human Patients

The human cohort used in this study consisted of 17 low-grade (LG) adenomas, 12 high-grade (HG) adenomas, and 75 adenocarcinomas (Appendix A). Biopsies (*n* = 30) and whole tumor specimens (*n* = 74) of patients who underwent either colonoscopy or surgical resection between 2004 and 2019 at the University Hospital Klinikum rechts der Isar Munich (MRI) of the Technical University of Munich, Germany were analyzed retrospectively. For 81 intestinal lesions, adjacent non-neoplastic intestinal mucosa was available on the same slide. Data was extracted from the Munich Cancer Registry as well as from hospital records. The use of human tissue was approved by the local ethics committee of the Technical University of Munich/Klinikum rechts der Isar (reference number: 506/17s).

### 2.3. Histology, Immunohistochemistry, and Slide Digitalization

Serial tissue sections of 2 µm thickness were cut from human and murine FFPE blocks using a rotating microtome (RM2245 Leica Biosystems, Wetzlar, Germany). Hematoxylin and eosin (H.E.) stainings were performed according to a standard protocol. Due to effective heparin absence, intestinal MMCs are sensitive to formalin fixation and thus cannot be detected by metachromatic staining methods, such as standard toluidine blue (T.B.) staining [14,15]. In order to confirm this, T.B. staining at pH 2.3 (Toluidine blue O, 1B-481.00010, Waldeck, Münster, Germany) was also performed (Appendix A).

For the identification of intestinal MMCs, IHC was applied. For each cohort, serial FFPE sections were cut gradually, and empty cuts were stored properly and as short as possible. Sectioning and immunohistochemical experiments were performed stepwise but in a timely manner between 01/2020 and 11/2021 on IHC autostainers (murine tissue on Leica Bond RX^m^, Leica, Wetzlar, Germany and human tissue on a Ventana BenchMark XT, Ventana Medical Systems, Tucson, AZ, US). Briefly, mouse serial sections were examined immunohistochemically by using primary antibodies for murine mast cell protease 1 (MCPT1) (mMCP-1, [RF6.1], a β-chymase [43]), mast cell tryptase (MCPT6) (TPSAB1; homologous to mMCP-6 [44]), granzyme B, and histamine. On human tissue, MC-Tryptase (MCT) [AA1], MC-Chymase (MCC) [CC1], CD117, and histamine antibodies were applied. Additionally, an anti-Ki-67 antibody was applied in order to visualize tumor cell proliferation. For each antibody, adequate positive and negative (tissue and reagent) controls were included. If true invasion of neoplastic cells was inconclusive on an H.E. slide, further sectioning, PAS reaction, and/or Pan-cytokeratin IHC were performed in order to obtain a secure diagnosis regarding tumor grade.

In order to gain first insights into the functional role of ieMMCs in intestinal neoplasia, we also performed Interleukin-5 (IL-5) (*n* = 28) and matrix metallopeptidase 9 (MMP-9/gelatinase B) (*n* = 19) on selected slides. Immunohistochemical double staining as a proof of principle, i.e., visualization of interepithelial located mast cells above the basal membrane, was also applied. A combination of collagen IV (for visualization of the basal lamina) and MCC or MCT (for visualization of MCs) was used. Antibody binding was visualized with a brown chromogen (3,3′-diaminobenzidine (DAB), Bond Polymer Refine Detection, DS9800, Leica Biosystems, Wetzlar, Germany) and a red chromogen (alkaline phosphatase, Bond Polymer Refine Red Detection, DS9390, Leica Biosystems, Wetzlar, Germany).

For details on the panel of primary antibodies used in this study, the corresponding working dilutions, and IHC protocols, see Appendix A.

After laboratory processing, slides were scanned in 40× magnification using a whole-slide brightfield scanner (Aperio AT2, Leica Biosystems, Wetzlar, Germany).

### 2.4. Histological Grading of Intestinal Neoplastic Lesions

A trainee pathologist (T.G.) and a board-certified veterinary pathologist (K.S.) independently evaluated H.E. slides. Intestinal lesions were classified according to the current state-of-the-art toxicopathological criteria for gastrointestinal lesions [30,45]. For a short overview of the applied intestinal lesion classification and grading scheme, see Appendix A. Murine lesions were graded as: atypical hyperplasia; low-grade adenoma; high-grade adenoma; or adenocarcinoma with an invasive component (Appendix A). Lesions were included only if connection to the tunica muscularis of the intestinal wall was retained. In total, 274 murine intestinal neoplastic lesions were annotated in tissue sections originating from 100 different mouse organisms of 12 different genotype groups (Appendix A). The different GEMMs were grouped according to (1.) their genotype; (2.) mode of tumor induction (genetic-, chemical-, or viral-based).

Human intestinal lesions were subtyped, graded, and staged by board-certified human surgical pathologists (C.M., M.J., W.W.) according to the current WHO classification of tumors of the colon [46].

### 2.5. Descriptive Semiquantitative Score of Mast Cell Infiltration

Semiquantitative scoring allows for robust group comparisons and inter- and intra-observer reproducibility of descriptive tissue data [47]. Therefore, a semiquantitative tissue score was established for determining occurrence of ieMMCs and lpMMCs in intestinal neoplastic lesions of mice and humans. An ordinal 5-tier-score (0 to 5 representing no (0) to high (5) ieMMC density) with species-specific cut-off values was set (Table 1 and Table 2; Appendix A). Validation and reproducibility of the applied morphologic scoring system were achieved by scoring selected slides through a second observer, who was a board-certified veterinary pathologist (inter-observer; K.S.), as well as by randomly scoring slides repeatedly by the first observer (intra-observer; T.G.). Pathologists were blinded to mouse model groups. Digitalized slides were scored by using the counter tool of the Aperio ImageScope x64 software (v.12.4.0.7018, Leica Biosystems Pathology Imaging, Wetzlar, Germany) on a standard monitor (Wacom Cintiq 22HD, Kazo, Saitama, Japan).

#### 2.5.1. Semiquantitative Score for Murine Tissue

In mice, IHC positive (^+^) ieMMCs, in terms of a brown DAB-precipitate for MC markers (MCPT1, granzyme B, MCPT6) and a morphologically addressable interepithelial location (Figure 1), were separately counted in MC hotspots (40× high-power field, hpf) of intra-tumoral and of tumor-adjacent normal tissue on the same slide. Slides were first screened under low power in order to identify areas with the highest MC density (MCD). High-power fields with the highest density of ieMMCs were defined as hotspots (Figure 2). MC counts in these hotspots were scored from 0 (no mast cells) to 5 (positive staining in >30 densely clustered mast cells/hotspot) (Table 1, Appendix A). If available on the slide, normal mucosa adjacent to the neoplastic lesion was also scored (*n* = 268).

Notably, ieMMC scores clearly differed between the apical and basal tumor area (lower third of the tumor next to the submucosa) or tumor center (if lesions were cut transversally), respectively. Thus, in a first step separate scores of apical tumor regions (luminal) vs. tumor basis/center (submucosal site) were applied (Figure 3). Finally, the apical MCPT1^+^ score (accounting for the highest MCD of the whole tumor) was considered to be representative.

Granzyme B^+^ and MCPT6^+^ ieMMC scores were determined for hotspots without differentiating between zones. The same score as described above was applied for granzyme B and MCPT6 (Table 1, Appendix A). Additionally, lpMMCs (MCPT6^+^) were counted and scored as described for ieMMCs (Appendix A).

#### 2.5.2. Semiquantitative Score for Human Tissue

In human lesions, in sharp contrast to murine lesions, hotspots of clustered ieMMCs were not present. In order to gain an overview of the entire human neoplastic tissue specimen, positive ieMMCs (MCC^+^, MCT^+^, CD117^+^), as defined above (Figure 1), were counted in 10 randomly selected 20× fields of neoplastic and also in matched adjacent normal colonic mucosa (if available on the same slide). Total ieMMC counts were scored from 0 (no positive cells) to 5 (dense (>30) MCs in ten 20× fields)) according to Table 2. In human specimens, lpMMCs and CTMCs were pooled as intratumoral stromal mast cells (ITSMCs), since a clear delineation according to cell size, shape, and location was not possible [48].

Intratumoral stromal mast cells (ITSMCs) located between neoplastic crypts (MCC^+^, MCT^+^) were scored from 0 (no MCs in ten 20× fields) to 5 (>200 lpMMCs in ten 20× fields) (Appendix A). In human specimens, congruent areas were evaluated for all applied MC IHC markers and scores.

### 2.6. Computer Assisted Analysis of Tumor Cell Proliferation (Ki-67)

Ki-67 index of whole murine tumors was assessed by means of a computer-assisted algorithm using QuPath version 0.3.0 for quantification [49]. Areas were annotated by using a pencil and touch monitor (Wacom Cintiq 22HD, Kazo, Saitama, Japan). The default set of parameters of the algorithm was modified according to the stain contrast and intensity of the scanned images. Cell segmentation was performed using the settings specified in Appendix A. Cell classification (tumor cells or stromal cells) was done after training an object classifier using ‘Random trees’ as a machine learning method [50]. Proliferation in terms of nuclear positive Ki-67 staining was calculated within the class “tumor cells” (% Ki-67^+^ cells/tumor cells).

### 2.7. Statistical Analyses

Statistics were performed using the SPSS Statistics v. 27 software (SPSS Inc., Chicago, IL, USA). Descriptive statistical tests (mean, standard deviation (SD), median, interquartile range (IQR)) were calculated according to standard methods.

Following the collection of semiquantitative ordinal scoring data as described above, non-parametric statistical analyses for various independent group comparisons (atypical hyperplasia; low-grade adenoma; high-grade adenoma; adenocarcinoma) were performed (one-way ANOVA Kruskal–Wallis test (KWT) with post-hoc Bonferroni multiple comparison test).

For comparison of non-normally distributed ieMMC scores within neoplastic and normal colonic compartments, a Mann-Whitney U (MWU) test was applied for statistical analysis. For all analyses, *p* < 0.05 was considered statistically significant.

## 3. Results

### 3.1. Comparative Histopathological Description and Characterization of Intestinal Mucosal Mast Cells

Overall, MCs were located above the epithelial basal membrane, between intestinal epithelial cells (ieMMCs), in murine as well as in human neoplastic intestinal mucosa (Figure 1). The morphological definition of ieMMCs was based on the study of Vogel et al., 2018 [17]. The interepithelial aspect of ieMMCs was clearly present in humans and mice (Figure 1), while immunohistochemical (granular) characteristics differed to some extent (Table 3).

In our comparative experimental pathology laboratory [51], a considerable number of intestinal lesions generated in GEMMs has been evaluated histologically. An intriguingly prominent accumulation of small round cells containing large brightly eosinophilic granules similar to MC granules—located in intestinal hyperplastic, adenomatous, or carcinomatous lesions—was evident in H.E. specimens. In the murine intestine, MCPT1 is a specific marker for ieMMCs according to Vogel et al., 2018 [17], which was also the case in our study. Additionally, murine ieMMCs were also positive for tryptase (MCPT6) (*n* = 248) and granzyme B (*n* = 271), but were negative for histamine (*n =* 37) (Appendix A).

In human tissue, MCT, MCC and CD117 were found to be appropriate markers for ieMMCs. In general, human ieMMCs showed positivity for MCT more frequently than for MCC (MC_TC_ included) and for CD117 (Table 4A). Human ieMMCs were consistently negative for histamine (*n =* 51) (Appendix A).

Tryptase-positive murine lpMMCs were clearly identified and were located in the intestinal lamina propria with an appearance that was slender and smaller than that of CTMCs (Appendix A). In human intestinal neoplasia, considering their larger size, more invasive nature, and complex tissue architecture, a clear distinction between lpMMCs and CTMCs in terms of cell size and shape could not be achieved. Due to this fact, in the human setting, both lpMMCs and CTMCs were grouped into intra-tumoral stromal mast cells (ITSMCs), defined as MCs located in the stroma between neoplastic crypts (Appendix A). Individual human CTMCs appeared to be unspecifically positive due to the presence of heparin within the MCs, causing ionic interactions with antibodies [52], while human ieMMCs were negative in negative reagent controls (Appendix A).

### 3.2. Human

#### 3.2.1. Interepithelial and Stromal Mast Cell Scores in Human Precursor Lesions, Adenocarcinomas, and Adjacent Normal Intestinal Mucosa

In human intestinal lesions (*n =* 104), intratumor ieMMC scores (MCT^+^, MCC^+^) decreased from precursor lesions (low-grade (LG) and high-grade (HG) adenoma) to adenocarcinoma (*p <* 0.0001, KWT) (Figure 4). Expression of CD117 (*n =* 51) revealed the same trend (*p =* 0.019, KWT) (Table 4A). Additionally, the presence of ieMMCs (MCT^+^, MCC^+^) was assessed in adjacent normal mucosa (*n =* 81) of LG and HG adenoma and adenocarcinoma. Scores for ieMMCs in adjacent mucosa were lowest for cases with HG adenoma, intermediate for patients with adenocarcinoma, and highest adjacent to LG adenoma (*p* = 0.047 (MCT) (Figure 5) and *p =* 0.018 (MCC), KWT) (Appendix A).

Human ITSMCs (MCT^+^ and MCC^+^), which were generally more abundant than ieMMCs, decreased from precursor lesions to adenocarcinoma (*p <* 0.001, KWT) (Appendix A). The same trend was evident for CD117^+^ ITSMCs (*p <* 0.001, KWT), whereas for histamine-positive ITSMCs, stromal MCD was generally reduced and lowest for LG adenomas, intermediate for adenocarcinomas, and highest for HG adenomas (*p =* 0.167, KWT) (Appendix A).

#### 3.2.2. General Comparison of ieMMC Scores in Normal Mucosa vs. (Pre-)Neoplastic Human Tissue

Generally, human ieMMC scores (MCT^+^, MCC^+^) were lower in neoplastic tissue compared to normal mucosa (*p <* 0.001, MWU test) (Figure 5) (Appendix A). In the adjacent mucosa, MCT^+^ ieMMCs were more frequently detected than MCC^+^ ieMMCs, which was in accordance with the protease content of ieMMCs located in neoplastic tissue. Thus, no phenotypic alteration with regard to mast cell proteases between normal and neoplastic intestinal mucosa was observed.

### 3.3. Mouse

#### 3.3.1. Interepithelial and Lamina Proprial Mucosal Mast Cell Scores in Murine Precursor Lesions, Adenocarcinomas, and Adjacent Intestinal Mucosa

Histological scores for MCPT1^+^, MCPT6^+^ and granzyme B^+^ ieMMCs showed that ieMMC density decreased from precursor lesions (atypical hyperplasia; LG; HG adenoma) to adenocarcinoma (*p <* 0.001) (Table 4) (Figure 6B–D). In contrast to this finding, ieMMC density in adjacent normal mucosa increased from precursor to carcinomatous lesions (*p* = 0.690) (Figure 5G).

In mice, ieMMCs were most prominent in the apical region of the neoplasia. Separate scores of apical tumor regions (luminal) vs. tumor basal/central regions (submucosal site) showed that scores were generally higher in the apical tumor (Figure 3).

LpMMCs were visualized by evaluating MCPT6^+^ cells located in the lamina propria (*n* = 248). Compared to the decreasing number of ieMMCs from precursor to carcinomatous lesions, lpMMC scores increased from precursors to carcinoma (*p =* 0.001) (Appendix A).

#### 3.3.2. General Comparison of ieMMC Scores in Normal Mucosa vs. (Pre-)Neoplastic Murine Tissue

Across all lesion groups, scores for ieMMCs were lower in the adjacent normal mucosa (*n =* 268) than in neoplastic tissue. Generally, numbers of MCPT1^+^ ieMMCs decreased from the apical to the basal/central tumor areas (Figure 3) and the normal mucosa (Figure 5). A MWU test revealed a significant difference between higher (pre-)neoplastic and lower normal adjacent mucosal ieMMC scores in murine intestinal lesions (*p =* 0.000) (Figure 5).

#### 3.3.3. Comparison of ieMMC Scores in Different Intestinal Carcinogenesis GEMMs and Intestinal Localizations

Scores for ieMMCs in the tumor area differed between the investigated GEMMs (*p <* 0.0001, KWT) (Figure 7A). For details on the GEMMs (pseudonymized) please refer to Appendix A. Considering all lesions regardless of grade and intestinal location, ieMMC scores assessed in lesions of BRAF mice (*n =* 20; median score 0.5) were generally low in comparison to other models (BRAF vs. βCAT *p* < 0.0001 (βCAT median score 3.5); BRAF vs. IE-CBM-deficient *p* < 0.001 (IE-CBM median score 3); BRAF vs. βCAT+IE-CBM-deficient *p* = 0.034 (βCAT+IE-CBM median score 2); BRAF vs. APC *p* = 0.075 (APC median score 2)) (Figure 7A). Additionally, in KRAS mice, ieMMC density was low (median score 1.5). However, increasing this specific cohort would be important to confirm this result (Figure 7A).

Scores for ieMMCs were generally high in lesions of the βCAT model (*n* = 26; median score 3.5) (βCAT vs. BRAF *p* < 0.0001; βCAT vs. complex models *p* < 0.0001 (complex median score 0); βCAT vs. WT *p* = 0.007 (WT median score 1); βCAT vs. APC *p* = 0.138) (Figure 7A).

We found no strong difference between the ieMMC scores of AOM-DSS-treated WT mice (*n* = 60; median score 1) and genetically modified mice (*p =* 0.563, MWU; data not shown). When lesions were separately analyzed by tumor grade, there was generally no clear evidence for a decrease in the occurrence of ieMMCs in the course of adenoma-carcinoma progression of each individual GEMM (data not shown) as it was observed in spontaneous human lesions (Figure 4). In virally induced GEMMs, where tumors were surgically induced by Adenovirus-Cre [34], ieMMC tumor infiltration was lower (median score = 0) than in AOM-DSS-induced (median score = 1.5) and endogenous (median score = 2) models (*p <* 0.001, KWT) (Figure 7B).

Most strikingly, murine ieMMC scores were generally higher in small intestinal lesions than in colonic neoplasia (*p* < 0.001, MWU) (Figure 8A). Decreasing ieMMC scores were found in the atypical hyperplasia–adenoma–carcinoma progression of murine colonic lesions (*p* < 0.001) (Figure 8B). In contrast, strong group differences between ieMMC scores in small intestinal lesions of different tumor grades were not evident (*p =* 0.222) (Figure 8C).

#### 3.3.4. Tumor Proliferation and First Functional Investigations of Murine Intestinal ieMMCs

In order to elucidate the correlation between tumor cell proliferation with the presence of MCs, we also assessed proliferation (Ki-67) of whole tumor areas and found that lower ieMMC scores were associated with a lower Ki-67 proliferative activity (*p* < 0.001, KWT). Tumor proliferation, as assessed by a high percentage of Ki-67^+^ tumor cells, was associated with an increased occurrence of ieMMCs (Figure 6E).

Aiming at elucidating the mechanistic effects of ieMMCs we also investigated the immunohistological presence of MMP-9 and IL-5 in the cohorts. Finally, a histomorphological correlation between the presence of ieMMCs in areas of intestinal neoplasia and MMP-9 or IL-5 immunoreactivity was not possible to infer. MMP-9^+^ cells did not colocalize with ieMMCs (Appendix A). Luminal intestinal crypt cells were often positive for IL-5, but IL-5 positivity did not correlate with ieMMC density (Appendix A).

### 3.4. Comparison of ieMMCs in the Adenoma-Carcinoma Sequence of Mice and Humans

IeMMCs were a common finding in intestinal neoplastic epithelium of mice (hotspots) and humans. Overall, the number of ieMMC decreased from LG to HG adenomas and to adenocarcinomas in both species. Importantly, ieMMC scores in human and murine colonic carcinomas were generally lower than in adenomas, suggesting that ieMMCs are immune pioneers in the colonic TME that enhance tumor progression and subsequently retreat in carcinomas. This trend was most evident in colonic lesions of different GEMMs (Figure 8B) and in human CRC (Figure 4), whereas in murine small intestinal lesions, ieMMC scores did not differ between carcinomas and LG adenomas (Figure 8C).

In contrast to mice, ITSMCs in human intestinal neoplastic lesions were generally more abundant and scores were highest in HG adenomas (Appendix A).

A difference between mice and humans was the decreased ieMMC score in murine normal tissue adjacent to neoplastic epithelium, whereas in humans, ieMMC density in adjacent normal tissue increased in comparison to neoplastic epithelium (Figure 5).

## 4. Discussion

The pathogenetic and clinically relevant role of MCs in inflammatory and allergic diseases as well as in mastocytosis is well-known and elucidated [53,54,55]. In humans, MCs are typed with respect to their granule content: tryptase (MC_T_); chymase (MC_C_); and tryptase- and chymase- (MC_TC_) positive MCs [56,57]. In mice, MCs are categorized based on their topographic location: connective tissue MCs (CTMCs); mucosal MCs (MMCs) [16,48,58]. Vogel et al., 2018 specified the murine MC classification by introducing the terms interepithelial mucosal MCs (ieMMCs) and lamina proprial mucosal MCs (lpMMCs) on one hand and connective tissue MCs (CTMCs) on the other hand [17]. These MC subtypes differ in their stored protease content and therefore in their biological effects [43]. In the murine intestine, ieMMCs express MCPT1 and MCPT2 (chymases), whereas lpMMCs and CTMCs express chymases and tryptases [17,59]. In the human intestinal mucosa, tryptase is the major enzyme of MMCs (MC_T_ or MC_TC_) [57,60]. However, specific knowledge of the enzymatic properties of human intestinal ieMMCs is still lacking.

Since Ehrlich’s (the “father of mast cells”) times, it has been known that MC numbers increase in tumors [3,61,62] and it is speculated that they might have ambiguous functions in tumorigenesis in both mice and humans [2,62]. On one hand, MCs are found to promote tumor angiogenesis [63] and metastasis [24], suggesting a potential oncogenic role. On the other hand, MCs contribute to the improvement of host immunity against neoplastic cells [7,8,64], suggesting a tumor-suppressive effect. Mast cells of the TME interact with other immune cells [65] and neoplastic cells [10]. Recent research has focused on elucidating the interaction of primary tumor cells with MCs [9,10,11,66] and new therapeutic approaches have relied on the identification of MCs as possible biomarkers [6,67] and modulators of MC-dependent tumor promoting mechanisms [11,68,69,70].

In this observational study, we focused on the comparative characterization and description of ieMMCs in intestinal cancer of mice and humans in order to further elucidate the role of MC subsets in the TME. We investigated the occurrence of MMCs, especially of the newly defined ieMMC subtype [17], along the adenoma-carcinoma sequence of intestinal carcinogenesis. For that, we performed a retrospective evaluation of a large cohort of human samples and several mouse models with a strong focus on inter-species comparison between humans and regularly used GEMMs of biomedical CRC research. Our study delivers three main findings: 

First, a prominent accumulation of ieMMCs in intestinal neoplastic lesions of mice and also humans was immunohistochemically identified. While in the former, ieMMCs were primarily arranged in hotspots, human ieMMCs were diffusely distributed within (pre-)neoplastic lesions. The motile MC is able to cross the basal lamina and to diffuse into the epithelium [20]. We immunohistologically verified MCPT1 (a β chymase [43]) as a specific ieMMC IHC marker in mice. In accordance with previous studies [59,71] we found that ieMMCs of murine intestinal lesions were, to a lower extent, MCPT6^+^ (MCT^+^), whereas no immunoreactivity of ieMMCs for MCT is described in intestinal helminth infection [17,72,73]. It is tempting to speculate that a switch in tryptase content might occur in tumor-infiltrating ieMMCs compared to ieMMCs which are related to helminth-induced immunopathology. Mast cell tryptase (MCT) is a well-known pro-angiogenic mediator [63,74] and it is strongly associated with protease-activated receptor 2 (PAR-2) expression on intestinal cells, thus contributing to tumor progression [9]. Accordingly, our data showed that an increased ieMMC score was generally associated with an increased level of tumor cell proliferation in murine intestinal tumors, being indicative of a proliferation-promoting impact of ieMMCs on intestinal cancer cells (Figure 6E).

In humans, few studies explicitly address ieMMCs as MCs located “within the epithelium”, most often in airway diseases [75,76]. We show for the first time that, comparable to murine colonic carcinogenesis, ieMMCs frequently occur in the course of human colorectal carcinogenesis. In contrast to mice, MCT^+^ ieMMCs were present even more frequently than MCC^+^ ieMMCs (including MC_TC_) (Figure 4). In accordance with our findings, Flores de los Rios et al., 2020 showed that MCT^+^ MCs were more abundant and more frequently degranulated in human metastatic CRC [24].

The second finding was that colonic ieMMC scores generally decreased in the course of adenoma-carcinoma progression in both species, whereas small intestinal ieMMC scores did not. In our study, various small and large intestinal cancer GEMMs were analyzed. In the majority of previous studies, grading of mouse tumors is obscure and not according to standard pathological guidelines (e.g., “polyps”). We graded murine intestinal lesions according to the current toxicopathological state-of-the-art grading [30,45], which is in line with the human WHO classification [46], and investigated the occurrence of ieMMCs with respect to this comparative classification of the different stages of intestinal carcinogenesis.

Overall, scores of murine ieMMCs decreased from atypical hyperplasia to carcinoma (Table 4B, Figure 6B–D). The accumulation of MCs in murine adenomatous lesions has been previously described [7,59,77]. Gounaris et al., 2007 state that murine polyps are regularly infiltrated with MCPT2^+^ MCs and term them intraepithelial MCs [59]. These are morphologically and most likely also functionally equivalent to the herein described MCPT1^+^ ieMMCs. Moreover, the authors show that polyp-associated MCs are essential for the onset of intestinal adenomas [59]. In a following study, Saadalla et al., 2018 favor an MC-linked multistep carcinogenesis of murine small intestinal cancer that is rather dependent on MC subtype than on MC density, linked to the finding that MCPT6^+^ and MCPT5^+^ CTMCs expanded in invasive tumors [77]. In our hotspot-based evaluation, we found a numerically significant decrease of intratumoral ieMMCs during adenoma-carcinoma progression (Table 4B). However, further investigations of the biological relevance need to be conducted in the future. These should specifically focus on the contribution of the ieMMC subtype to intestinal carcinogenesis in variable molecular conditions, which are strongly dependent on the employed GEMM.

Interestingly, in colorectal lesions of mice, ieMMC density clearly decreased during the progressive sequence of carcinogenesis; however, ieMMC scores of small intestinal lesions were generally higher and did not decrease from low-grade adenoma to carcinoma (Figure 8). Our data are in agreement with the notion that murine small intestinal tumors have a different immunological milieu than colonic tumors and that MCs might therefore affect adenoma-carcinoma progression differently in the small intestine [77]. It is important to mention that a non-negligible proportion of GEMMs in CRC research (e.g., *Apc*-mutated) develop tumors predominantly in the small intestine [19,31], which is in sharp contrast to the human situation [78]. This indicates the importance of employing adequate mouse models, especially when it comes to the investigation of the immune microenvironment and possible immunotherapeutic strategies.

Our third major finding was that intratumoral ieMMC scores differed between variable GEMMs. Median score was low in WT, indicating an enhancement of ieMMC–tumor cell interaction related to specifically induced molecular carcinogenic changes. Scores were highest in βCAT, whilst they were lowest in the BRAF cohort and also low in the KRAS mouse model (Figure 7A). Via tryptase release, ieMMCs are able to activate PAR-2 on intestinal cells, which is connected to major G-protein-mediated signaling pathways, enhancing the Ras-Raf-MEK-ERK (MAPK) signal transduction cascade [9,11]. Since this cascade is already constitutively activated in *Braf^V637E^*- and *Kras^G12D^*-driven intestinal cancer models [33,36], the enhancement of MAPK signaling via MCs [11] might play a minor role in MAPK-driven intestinal cancer and ieMMC numbers remain low. It has previously been shown that Wnt/β-catenin signaling promotes maturation of MCs and their pro-tumorigenic activity in intestinal cancer [71,79]. However, it still remains unclear which factors induced by strong Wnt/β-catenin signaling in the intestinal tumor cell led to this immense MC migration, as it was seen predominantly in the βCAT and also APC model group (Figure 7A). Vascular endothelial growth factor (*VEGF*) is one amongst several upregulated target genes of Wnt signaling [80,81] and it is known that VEGF acts as a chemoattractant for MCs [82]. Recent research focuses on the molecular interaction of human colon cancer and MCs [10,11,66]. However, the concrete underlying mechanisms, enhanced by a close spatial relationship between the cancer cell and adjoining ieMMC, remain elusive. Further research is imperative in order to elucidate the possible biological pathways through which cancer cells drive MC migration, differentiation, and mechanisms and vice versa.

Finally, we hypothesize that the ieMMC subset is a particularly important component in the onset and progression of intestinal neoplasia in mice and in humans. Due to the spatial proximity of ieMMCs to the adjoining neoplastic epithelial cells, their influence on cancer cells is particularly interesting. We suggest that the functional role of ieMMCs in the TME differs from stromal MCs in murine and human CRC. Moreover, it will be tempting to investigate, if the prognostic significance of ieMMC and stromal mast cell density differs in humans and if ieMMC density itself could serve as an “ideal biomarker” for human CRC [67].

Advanced functional studies in MC-deficient mouse models [83,84,85,86] and second-generation GEMMs for intestinal cancer (e.g., tumor initiation plus Cre-mediated immune cell manipulation) will be required for ultimately determining the role of the MC subpopulations in intestinal cancer. For this reason, it is important to consider for instance that ablation of *Mcpt5* will not target ieMMCs. It will be vital to employ GEMMs, in which the efficient and specific targeting of all, or of one specific, MC subtypes is assured [85,86], considering the great relevance of MC heterogeneity in intestinal cancer. Furthermore, comparative studies focusing on potential differences in ieMMC and stromal MC occurrence between human samples and genetically similar specimens derived from murine models (e.g., *BRAF*-, *KRAS*-mutated tumors) combined with the prognostic assessment of MCs in human CRC should be conducted in the future. In addition, it is foreseeable that targeted therapies specific for MCs could be further explored in this context, thus offering another angle through which CRC could be treated [13].

## 5. Conclusions

We show that ieMMCs in murine and human intestinal lesions (FFPE) are reliably detectable using IHC markers such as MCC (MCTP1) and MCT (MCPT6). Interepithelial mucosal mast cells infiltrate the intestinal (pre-)neoplastic epithelium, and their numbers decrease along the adenoma-carcinoma progression of colonic tumors in mice and humans, whilst those of ieMMCs in murine small intestinal neoplasia as well as stromal MCs in both species do not. Therefore, we support a potentially different functional role of ieMMCs in comparison to stromal MCs during CRC progression. Given the obvious differences in the population dynamics of ieMMCs in (pre-)neoplasia of murine small and large intestine and also in different GEMMs for intestinal cancer, we strongly emphasize the need to employ adequate mouse models, especially when investigating the role of MCs within the immunologic milieu of CRC.

Mast cells are heterogeneous, multifaceted immune cells and deciphering their role and interaction with other immune cells and the adjacent epithelial cells is of the utmost importance. Increased knowledge of the MC role in the intestinal TME and of the interactions of ieMMCs and cancer cells will contribute to the improvement of strategies for curing CRC.

## Figures and Tables

**Figure 1 cancers-14-02248-f001:**
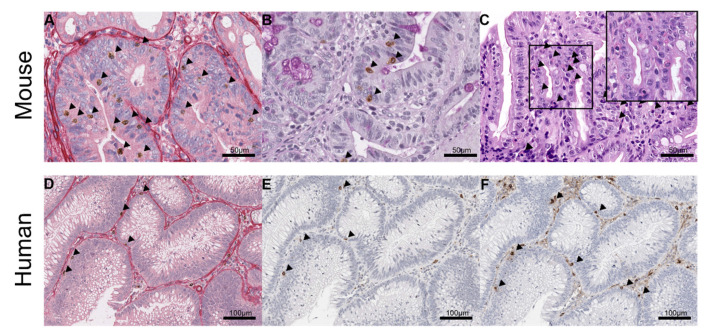
Identification of interepithelial mucosal mast cells (ieMMCs; arrowheads) in murine (**A**–**C**) and human (**D**–**F**) intestine. (**A**) Immunohistochemical sequential double staining of basal lamina (collagen IV) and ieMMCs (MCPT1) clearly reveals interepithelial localization of murine ieMMCs; (**B**) PAS reaction and ieMMC IHC (MCPT1) confirms interepithelial location of MCPT1^+^ cells; (**C**) Prominent accumulation of small, round cells containing large, brightly eosinophilic granules located in intestinal hyperplastic epithelium of a hematoxylin-eosin (H.E.) section; (**D**) Immunohistochemical sequential double staining of basal lamina (collagen IV) and ieMMCs (MC-Chymase; MCC) clearly reveals interepithelial localization of human ieMMCs; (**E**) Serial sections of MCC and; (**F**) MC-Tryptase (MCT) immunohistochemistry (IHC) in a human intestinal adenoma.

**Figure 2 cancers-14-02248-f002:**
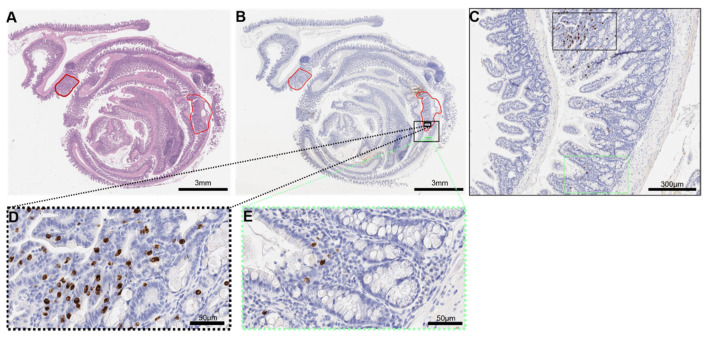
Analysis of ieMMCs in murine colon rolls. (**A**) Intestinal lesions were graded and annotated (red line) in an H.E. section; (**B**) Accordingly, a serial section of MCPT1 IHC was annotated; (**C**) Hotspots of MCPT1^+^ mast cells (MCs) were identified in neoplastic mucosa (black frame) and normal mucosa (green frame); (**D**) Mast cell numbers were counted in one 40× field (high-power field, hpf) of the respective hotspot. A semiquantitative score was applied (Appendix A) (Table 1); (**E**) In the adjacent normal mucosa, MC numbers were counted and scored accordingly.

**Figure 3 cancers-14-02248-f003:**
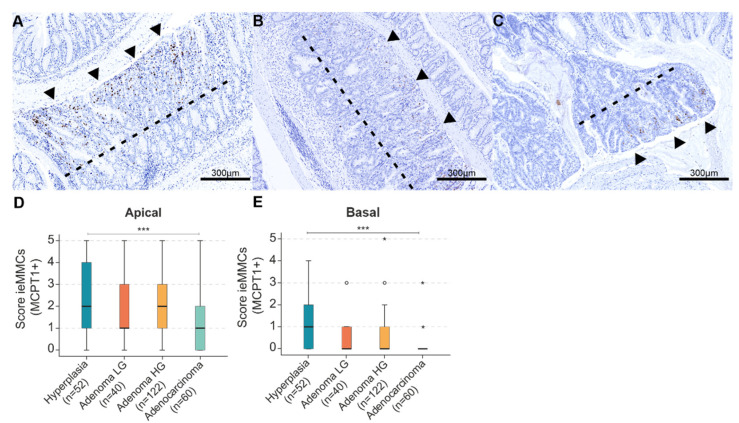
(**A**–**C**) In murine intestinal neoplasia (*n* = 274), MCPT1^+^ ieMMC were generally more abundant in the apical two thirds of the lesion (luminal site; arrowheads) compared to the basal, lower third of tumor tissue (submucosal site). The dotted line divides the different zones; (**D**–**E**) Generally, ieMMC scores were higher in the apical (**D**) than in the basal (**E**) areas of intestinal neoplastic lesions; (**D**) Kruskal–Wallis test (KWT) of ieMMC scores in the apical aspects of tumors during adenoma-carcinoma progression. Scores for ieMMC were higher in precursors (atypical hyperplasia and low- and high-grade adenomas) than in carcinomatous lesions (KWT *** *p* < 0.001); (**E**) KWT of ieMMC scores in the basal aspects of tumors during adenoma-carcinoma progression (KWT *** *p* < 0.001). * = extreme statistical outliers; ° = mild statistical outliers.

**Figure 4 cancers-14-02248-f004:**
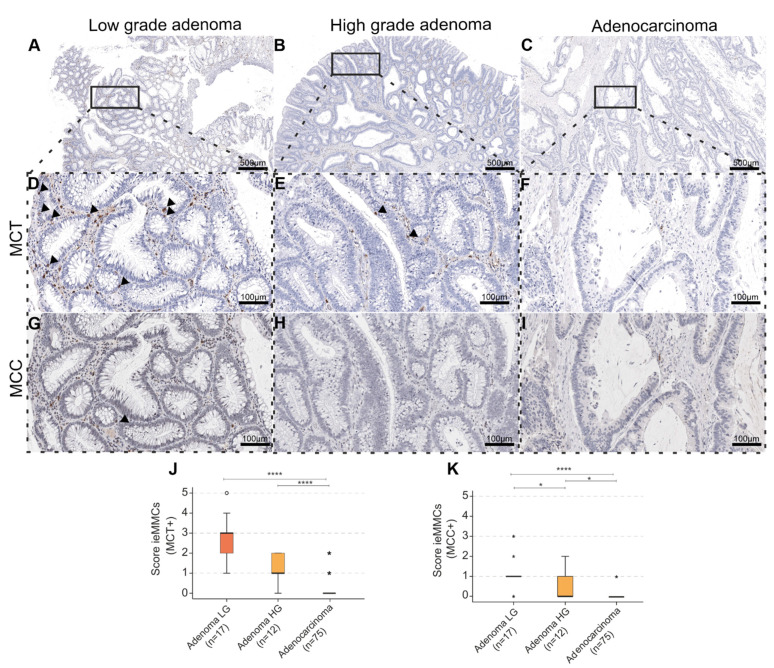
(**A**–**I**) In the context of human adenoma-carcinoma progression (**A**–**C**), ieMMCs (arrowheads) positive for MCT (**D**–**F**) and MCC (**G**–**I**) decreased from low-grade (LG) to high-grade (HG) adenoma and adenocarcinoma (ACA) (serial sections); (**J**,**K**) Generally, median scores for ieMMCs in adenomas were higher for MCT (KWT, pairwise comparison of ACA-LG **** *p* < 0.0001; ACA-HG **** *p* < 0.0001) (**J**); than for MCC (KWT, pairwise comparison of LG-HG * *p* = 0.022; HG-ACA **p* = 0.017; LG-ACA **** *p* < 0.0001) (**K**). * = extreme statistical outliers; ° = mild statistical outliers.

**Figure 5 cancers-14-02248-f005:**
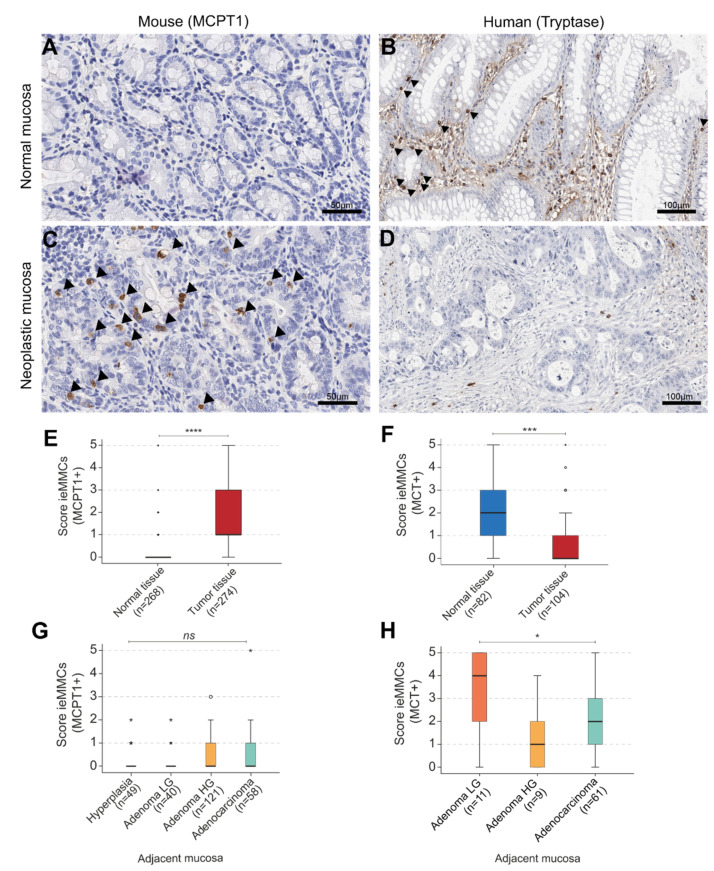
(**A**–**D**) Interepithelial mucosal mast cell (ieMMC) (arrowheads) in normal vs. neoplastic colonic mucosa of murine (**A**,**C**) and human (**B**,**D**) intestine; (**E**) Generally, scores of ieMMCs were higher in (pre-)neoplastic lesions of mice (Mann-Whitney U test (MWU), **** *p* = 0.000); (**F**) Contradictory, human ieMMC scores were higher in normal adjacent tissue (MWU, *** *p* < 0.001); (**G**) IeMMCs in the adjacent normal mucosa of mice increased in the course of adenoma-carcinoma progression (KWT, *p* = 0.690, not significant (ns)); (**H**) In humans, density of ieMMCs in adjacent normal mucosa decreased from LG adenoma to carcinoma to HG adenoma (KWT, * *p* = 0.047). * = extreme statistical outliers; ° = mild statistical outliers.

**Figure 6 cancers-14-02248-f006:**
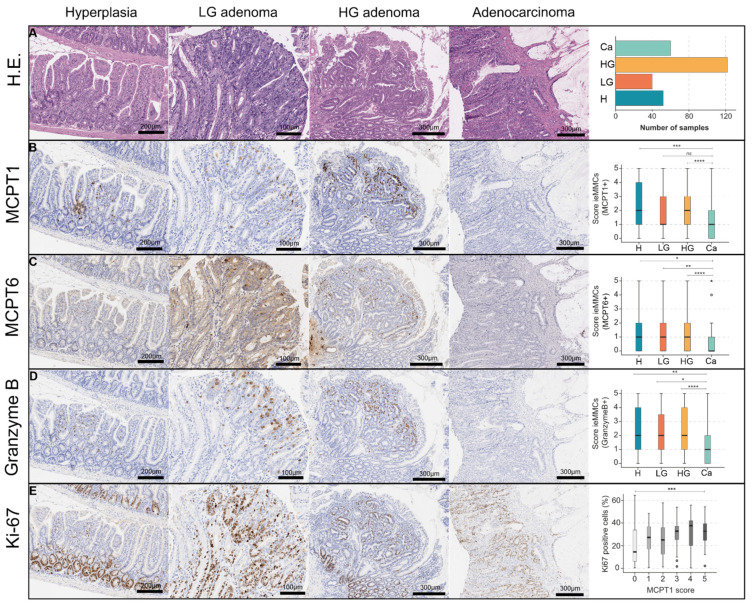
(**A**–**D**) In the context of murine adenoma-carcinoma progression (**A**) ieMMC scores generally decreased from low-grade adenoma to carcinoma (KWT, pairwise comparisons, * *p* < 0.05; ** *p* < 0.01; *** *p* < 0.001; **** *p* < 0.0001; not significant (ns) *p* > 0.05). IeMMCs were identifiable using MCPT1 (**B**), MCPT6 (**C**), and granzyme B (**D**) IHC; (**E**) Higher ieMMC scores were connected to a higher proliferation in terms of % Ki-67^+^ tumor cells in a neoplastic lesion (*n* = 271) (KWT, *** *p* < 0.001). * = extreme statistical outliers; ° = mild statistical outliers.

**Figure 7 cancers-14-02248-f007:**
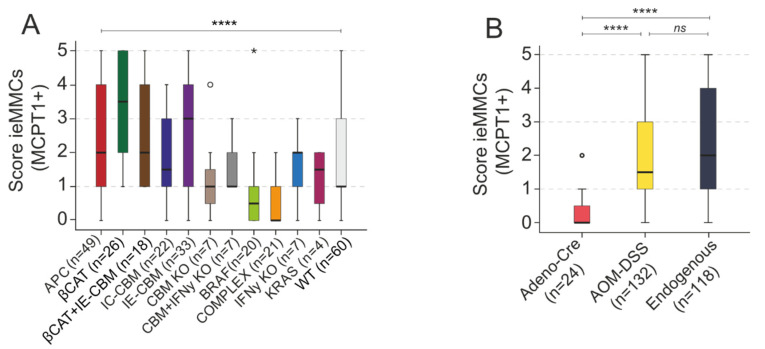
(**A**,**B**) Scores for ieMMCs over all tumor grades were compared according to genotype groups (KWT, **** *p* < 0.0001). For details on the pseudonymized genetically engineered mouse groups, please refer to Appendix A (**A**); additionally, group comparison according to mode of tumor induction was performed (KWT, *p* < 0.0001) (**B**). (**A**) Scores assessed in lesions of *Braf^V637E^* mice (BRAF) (*n =* 20; median score 0.5) were generally low, while scores were highest in lesions of the *Catnb^Δex3^* model (βCAT) (*n* = 26; median score 3.5); (**B**) Scores in Adeno-Cre-induced tumors were lower than in AOM-DSS-induced (KWT, pairwise comparison **** *p* < 0.0001) and in endogenous models’ scores (KWT, pairwise comparison **** *p* < 0.0001). There was no statistically significant effect between AOM-DSS and endogenous models (*p* = 0.636; not significant (ns)). * = extreme statistical outliers; ° = mild statistical outliers.

**Figure 8 cancers-14-02248-f008:**
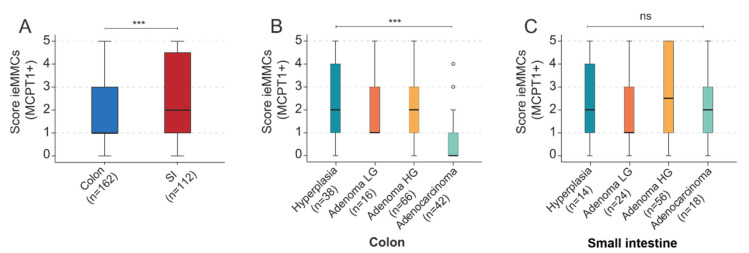
(**A**) Scores for ieMMCs were generally lower in murine colonic tumors (*n* = 162) than in murine small intestinal tumors (*n* = 112) (KWT, *** *p* < 0.001); (**B**) Scores for ieMMCs decreased from atypical hyperplasia to carcinoma in the colon (KWT, *** *p* < 0.001); (**C**) However, in the small intestine scores for ieMMCs did not decrease from precursors to invasive neoplasia (KWT, *p* = 0.222, not significant (ns)). ° = mild statistical outliers.

**Table 1 cancers-14-02248-t001:** Semiquantitative evaluation of murine interepithelial mucosal mast cells (ieMMCs).

Score	Description
0	No positive ieMMCs in the evaluated area
1	Positive staining of single ieMMCs (≤5/hpf ^1^) per hotspot ^2^
2	Positive staining of few diffuse ieMMCs (≤10/hpf) per hotspot
3	Positive staining of some diffuse ieMMCs (≤20/hpf) per hotspot
4	Positive staining of many diffusely distributed or possibly clustered ieMMCs (≤30/hpf) per hotspot
5	Positive staining of many ieMMCs (>30/hpf) and dense clustering per hotspot

^1^ high-power field (hpf; 40× field; Aperio ImageScope x64 v.12.4.0.7018); ^2^ one hotspot (= hpf with the highest MC density per tumor) was counted and scored; the number of positive ieMMCs was considered in two hotspots (one hotspot of tumor-adjacent normal tissue and one hotspot of the neoplastic area).

**Table 2 cancers-14-02248-t002:** Semiquantitative evaluation of human ieMMCs.

Score	Description
0	No positive ieMMCs in ten × 20 fields ^1^
1	Single positive (≤5) ieMMCs in ten × 20 fields
2	Few positive (≤10) ieMMCs in ten × 20 fields
3	Moderate number of positive (≤20) ieMMCs in ten × 20 fields
4	Many (≤30) ieMMCs in ten × 20 fields
5	Dense (>30) ieMMCs in ten × 20 fields

^1^ Ten × 20 fields (Aperio ImageScope x64 v.12.4.0.7018) were randomly selected throughout the neoplastic and normal mucosal area and separately scored.

**Table 3 cancers-14-02248-t003:** Reactivity for MC subtypes for each of the primary antibodies.

Marker	ieMMC ^1^ (Mouse)	lpMMC ^2^ (Mouse)	ieMMC (Human)	ITSMC ^3^ (Human)
MCPT1 (mMCP-1) ^4^	+	-	-	-
Anti-human MC-Chymase	n/a ^6^	n/a	+	+
Anti-human MC-Tryptase	n/a	n/a	+	+
TPSAB1 (mMCP-6) ^5^	+	+	+	+
Histamine	-	-	-	+
Granzyme B	+	+	n/a	n/a
CD117 (c-kit)	n/a	n/a	+	+

A “+” sign indicates that the respective mast cell subtype expresses the marker and a “-” sign indicates that the subtype does not express the marker. ^1^ Interepithelial mucosal mast cell (ieMMC); ^2^ lamina proprial mucosal mast cell (lpMMC); ^3^ intratumoral stromal mast cell (ITSMC); ^4^ mast cell protease 1 (MCPT1); murine mast cell protease 1 (mMCP-1); ^5^ tryptase alpha-1 and beta-1 (TPSAB1); murine mast cell protease 6 (mMCP-6); ^6^ not available (n/a) as markers were not assessed in or not appropriate for the respective species (n/a).

**Table 4 cancers-14-02248-t004:** (**A**) Interepithelial mucosal mast cell (ieMMC) counts (mean) in ten × 20 fields and scores (median) of human intratumoral intestinal mucosa (MCT, MCC, CD117). (**B**) Interepithelial mucosal mast cell (ieMMC) scores (median) of murine intratumoral and adjacent intestinal mucosa.

(**A**)				
**Lesion**	** *n* **	**ieMMC Tumor Count (mean ± SD ^1^) MCT ^2^**		**ieMMC Score** **(median ± IQR ^3^) MCT ^2^**
Low-grade adenoma	17	14.35 ± 11.784		3 ± 2
High-grade adenoma	12	4.17 ± 3.040		1 ± 1
Adenocarcinoma	75	0.47 ± 1.605		0 ± 0
*p* (KWT ^4^)	104	< 0.0001		
**Lesion**	** *n* **	**ieMMC tumor count (mean ± SD) MCC** ^4^		**ieMMC score** **(median ± IQR) MCC ^4^**
Low-grade adenoma	17	4 ± 4.287		1 ± 1
High-grade adenoma	12	1.25 ± 2.137		0 ± 1
Adenocarcinoma	75	0.16 ± 0.698		0 ± 0
*p* (KWT)	104	< 0.0001		
**Lesion**	** *n* **	**ieMMC tumor count (mean ± SD) CD117**		**ieMMC score** **(median ± IQR) CD117**
Low-grade adenoma	17	3.29 ± 9.841		0 ± 1
High-grade adenoma	12	0.67 ± 1.371		0 ± 1
Adenocarcinoma	22	0.05 ± 0.213		0 ± 0
*p* (KWT)	51	0.019		
**(B)**				
**Lesion**	** *n* **	**ieMMC score ^5^** **(median ± IQR)** **MCPT1 ^6^**	** *n* **	**ieMMC score** **(median ± IQR) MCPT6 ^7^**
Hyperplasia	52	2 ± 3	42	1 ± 2
Low-grade adenoma	40	1 ± 2	34	1 ± 2
High-grade adenoma	122	2 ± 2.25	115	1 ± 2
Adenocarcinoma	60	1 ± 2	57	0 ± 1
*p* (KWT)	274	< 0.001	248	< 0.001
**Lesion**	** *n* **	**ieMMC score** **(median ± IQR)** **granzyme B**	** *n* **	**ieMMC score** **(median ± IQR) MCPT1** **(adjacent mucosa)**
Hyperplasia	52	2 ± 3	49	0 ± 0
Low-grade adenoma	40	2 ± 2.75	40	0 ± 0
High-grade adenoma	121	2 ± 3	121	0 ± 1
Adenocarcinoma	58	1 ± 2	58	0 ± 1
*p* (KWT)	271	< 0.001	268	0.69

^1^ Standard deviation (SD); ^2^ mast cell tryptase (MCT); ^3^ interquartile range (IQR); ^4^ Kruskal–Wallis test (KWT), *p* < 0.05 statistically significant. ^4^ mast cell chymase (MCC). ^5^ In mice, the hotspot with the highest ieMMC score throughout the whole tumor was considered to be representative; ieMMCs positive for ^6^ mast cell protease 1 (MCPT1); ^7^ mast cell protease 6 (MCPT6).

## Data Availability

The raw data of the results presented in this study are available upon request from the corresponding author.

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
