# Peer review of "Comparative Study of the Role of Interepithelial Mucosal Mast Cells in the Context of Intestinal Adenoma-Carcinoma Progression"

_cancers, 2022, doi:10.3390/cancers14092248_

Round 1

Reviewer 1 Report

In this manuscript, the authors provide a thorough evaluation of mast cells in both human colorectal cancer specimens. The work is descriptive. There seems to be significantly different biology between the human and murine models. Nevertheless, the work could serve as a valuable resource for mast cell biologists, and the authors should be commended for their careful analysis, clear figures, and comprehensive discussion. Overall, I am supportive of publication with only grammatical editing.

Ex. “We found a compliant decrease of ieMMCs from colonic low grade adenomas to carcinomas..” Please replace the word compliant. There are similar suboptimal adjective word choices throughout the introduction and conclusion which detract from the science.

Figure 1-8 No major concerns. Of note, Fig 3E label should read Basal not Basis.

Table 4B, while I understand this may be numerically significant, is this biologically significant?

Author Response

In this manuscript, the authors provide a thorough evaluation of mast cells in both human colorectal cancer specimens. The work is descriptive. There seems to be significantly different biology between the human and murine models. Nevertheless, the work could serve as a valuable resource for mast cell biologists, and the authors should be commended for their careful analysis, clear figures, and comprehensive discussion. Overall, I am supportive of publication with only grammatical editing.
Response: We would like to thank the respected reviewer for these kind and supporting words.

  • “We found a compliant decrease of ieMMCs from colonic low grade adenomas to carcinomas..” Please replace the word compliant. There are similar suboptimal adjective word choices throughout the introduction and conclusion which detract from the science.

Response: Thanks for this suggestion and for the careful read. We adapted the manuscript accordingly. The word compliant (line 44) was removed and further suboptimal choices of adjectives and adverbs were replaced as well (e.g. line 94, line 103, line 114). Moreover, the conclusion part was revised thoroughly.

  • Figure 1-8 No major concerns. Of note, Fig 3E label should read Basal not Basis.
    Response: Thank your for your kind feedback and the advice regarding Figure 3. Figure 3E was adapted accordingly. Additionally, Fig. 7A was corrected (β-symbol instead of “B” and “WT instead of “Wt”).
  • Table 4B, while I understand this may be numerically significant, is this biologically significant?
    Response: Thank you for this solid objection. Further functional experiments are definitely needed to confirm the biological relevance of our descriptive immunohistological study. We added the following to the discussion part (starting from Line 632):
    “In our hotspot based evaluation, we found a numerically significant decrease of intratumoral ieMMCs during adenoma-carcinoma progression (Table 4 B). However, further investigations of the biological relevance need to be conducted in the future. These should focus specifically on the contribution of the ieMMC subtype to intestinal carcinogenesis in variable molecular conditions, which are strongly dependent on the employed GEMM.”

Reviewer 2 Report

This is an interesting article concerning the Role of Interepithelial Mucosal Mast Cells in the Context of Intestinal Adenoma-Carcinoma Progression 

I have the following comments:

The introduction is well developed.  

Moderate English-revision needed (some typos in the text. Ex. tissueu instead of tissues). 

Please add the period of the study 

The discussion should be shortened as well as improved by adding a deeper comparison with the literature. 

What are the clinical implications of the study?

What are the future perspectives?

Authors should highlight the clinical significance of the study by pointing out what it adds to the literature

Some articles may be considered and added in the introduction or discussion for completion 

Mastocytosis-A Review of Disease Spectrum with Imaging Correlation. Cancers (Basel). 2021 Oct 12;13(20):5102. doi: 10.3390/cancers13205102. PMID: 34680251; PMCID: PMC8533777.

Mast Cells, microRNAs and Others: The Role of Translational Research on Colorectal Cancer in the Forthcoming Era of Precision Medicine. J Clin Med. 2020 Sep 3;9(9):2852. doi: 10.3390/jcm9092852

Author Response

Response to Reviewer 2 Comments

This is an interesting article concerning the Role of Interepithelial Mucosal Mast Cells in the Context of Intestinal Adenoma-Carcinoma Progression 

Response: We would like to thank the respected reviewer for these kind and supporting words, the feedback and constructive comments.

I have the following comments:

  • The introduction is well developed.  
    Response: Thank you very much for the kind feedback.

  • Moderate English-revision needed (some typos in the text. Ex. tissueu instead of tissues). 
    Response: Thank you for the careful read. The text of the manuscript was revised thoroughly and all typos were corrected (e.g. line 48 adequate, line 226 occurrence, line 171 tissue). Moreover, another spelling-check was carried out and word spelling is now according to American English (e.g. line 126 analyzed; line 153 harboring).

  • Please add the period of the study.
    Response: Thank you for this suggestion. We added the period of the human cohort collection (2004-2019), as well as the period of the generation of investigated mouse models (2008-2020) and also the period of performed IHC experiments (01/2020-11/2021) to the materials and methods section (line 127, line 166, line 185).
    Mice were generated/sacrificed in 2009, 2010, 2016, 2017, 2018, 2019, 2020 and 2021. Four human cases date back to 2004, one case to 2007, however, the vast majority of investigated human cases is from the years 2012-2019.
    All FFPE blocks were stored properly (dark, dry and cool). Of note, serial tissue sections and IHC experiments were carried out gradually and in a timely manner. Empty cuts were stored as short as possible and in a dry, dark and cool pathology archive cabinet in order to avoid loss of antigenicity.
    Immunoreactivity of each FFPE specimen was assured by aligning staining results to matched FFPE positive control specimens. If available on slide, internal positive control reactions were also used.

  • The discussion should be shortened as well as improved by adding a deeper comparison with the literature. What are the clinical implications of the study? What are the future perspectives?
    Authors should highlight the clinical significance of the study by pointing out what it adds to the literature. Some articles may be considered and added in the introduction or discussion for completion:
    Mastocytosis-A Review of Disease Spectrum with Imaging Correlation. Cancers (Basel). 2021 Oct 12;13(20):5102. doi: 10.3390/cancers13205102. PMID: 34680251; PMCID: PMC8533777.
    Mast Cells, microRNAs and Others: The Role of Translational Research on Colorectal Cancer in the Forthcoming Era of Precision Medicine. J Clin Med. 2020 Sep 3;9(9):2852. doi: 10.3390/jcm9092852
    Response: Thanks for the feedback. The discussion part was shortened (e.g. Lines 591-597 and lines 689-697 were removed) as well as improved according to your suggestions. The suggested papers were also included. The addition to the literature was pointed out.

    We added a deeper comparison with the literature by the following modifications:

Line 543: “The pathogenetic and clinical relevant role of MCs in inflammatory and allergic diseases as well as in mastocytosis is well-known and elucidated [53-55]“ (Elsaiey et al. included).

Line 551: A deeper review of the MC proteases in mice vs. humans  was added: “These MC subtypes differ in their stored protease content and therefore in their biological effects [43]. In the murine intestine, ieMMCs express MCPT1 and MCPT2 (chymases), whereas lpMMCs and CTMCs express chymases and tryptases [17,59]. In the human intestinal mucosa, tryptase is the major enzyme of MMCs (MCT or MCTC) [57,60]. However, specific knowledge of the enzymatic properties of human intestinal ieMMCs is still lacking.”

Line 564: “Recent research has focused on elucidating the interaction of primary tumor cells with MCs [9-11,66] and new therapeutic approaches have relied on the identification of MCs as possible biomarker [6,67] and modulation of MC-dependent tumor promoting mechanisms [11,68-70]“
(included Sammarco et al. [67] and added an additional Reference here (Ammendola et al. 2016) [70]).

Line 590: Added another Reference [74] in order to verify the pro-angiogenic properties of MCT.

And we pointed out what we add to the literature:

Line 602: “We show for the first time that, comparable to murine colonic carcinogenesis, ieMMCs frequently occur in the course of human colorectal carcinogenesis.”

Moreover, our study aims at raising awareness for similarities, but also differences and limitations of mast cell populations in murine and human CRC, indicating the need for employing adequate mouse models and standardized IHC methodology.

Furthermore, the clinical relevance and future perspectives were substantiated:

Line 674: “ Finally, we hypothesize that particularly the ieMMC subset is an important component for onset and progression of intestinal neoplasia in mice and in humans. Due to the spatial proximity of ieMMCs to the adjoining neoplastic epithelial cells, their influence on cancer cells is particularly interesting. We suggest that the functional role of ieMMCs in the TME differs from stromal MCs in murine and human CRC. Moreover, it will be tempting to investigate, if the prognostic significance of ieMMC and stromal mast cell density differs in humans and if ieMMC density itself could serve as an “ideal biomarker” for human CRC [67].”

  • Are the conclusions supported by the results? Must be improved.

Response: The conclusion (Line 712-728) part was revised and improved:

“We show that ieMMCs in murine and human intestinal lesions (FFPE) are reliably detectable using IHC markers such as MCC (MCTP1) and MCT (MCPT6). Interepithelial mucosal mast cells infiltrate the intestinal (pre-)neoplastic epithelium and their numbers decrease along the adenoma-carcinoma progression of colonic tumors of mice and humans, whilst ieMMCs in murine small intestinal neoplasia as well as stromal MCs in both species, do not. Therefore, we support a potentially different functional role of ieMMCs in comparison to stromal MCs during CRC progression. Given the obvious differences in the population dynamics of ieMMCs in (pre-)neoplasia of murine small and large intestine and also in different GEMMs for intestinal cancer, we strongly emphasize the need to employ adequate mouse models, especially when investigating the role of MCs within the immunologic milieu of CRC.

Mast cells are heterogeneous, multifaceted immune cells and deciphering their role and interaction with other immune cells and the adjacent epithelial cells is of the utmost importance. Increased knowledge of the MC role in the intestinal TME and of the interactions of ieMMCs and cancer cells will contribute to the improvement of strategies for curing CRC.”
